# Type I Dentin Dysplasia: The Literature Review and Case Report of a Family Affected by Misrecognition and Late Diagnosis

**DOI:** 10.3390/medicina59081477

**Published:** 2023-08-17

**Authors:** Alessandra Putrino, Martina Caputo, Angela Galeotti, Enrico Marinelli, Simona Zaami

**Affiliations:** 1Department of Anatomical, Histological, Forensic and Orthopedic Sciences, Sapienza University of Rome, 00161 Rome, Italy; alessandra.putrino@uniroma1.it (A.P.); simona.zaami@uniroma1.it (S.Z.); 2Bambino Gesù Children’s Hospital, IRCCS, 00165 Rome, Italy; angela.galeotti@opbg.net; 3Department of Medico-Surgical Sciences and Biotechnologies, Sapienza University of Rome, 04100 Latina, Italy; enrico.marinelli@uniroma1.it

**Keywords:** abnormalities, dentine dysplasia, rootless teeth, shortened roots, litigation, malpractice, late diagnosis

## Abstract

*Background and Objectives*: Type I dentin dysplasia (DD-I) is a rare genetic disorder with autosomal dominant or recessive inheritance at risk of late or long-misunderstood diagnosis because the teeth, compared to other degenerative dentin diseases, do not have coronal defects and/or alterations but only at the root level (absent, conical, pointed roots, and obliterated pulp canals). The first radiographic suspicion often occurs only in case of sudden mobility and/or abscesses of the permanent teeth. Genetic tests confirm the diagnosis. *Case Presentation*: This case report describes the oral and radiographic characteristics of two siblings, 12 and 10 years old, a male and a female, at an early age affected by DD-I, whose diagnosis was made for a first orthodontic visit. The father and the older child had already undergone dental and orthodontic treatments, respectively, without the disease being suspected by the dentist. *Results*: Genetic tests support the diagnosis of DD-I. Following the diagnosis, the patients began a process of close periodic checks every 3–4 months to monitor their situation. The male child lost upper lateral incisors, which were then replaced with a light nylon removable prosthesis. *Conclusions*: The ability to recognize the radiographic features characteristic of DD-I is very important to avoid prejudicial diagnostic delays and to be able to plan the long-term treatment of these patients better, especially when the pathology was primarily misrecognized in the family.

## 1. Introduction

Originally, in Shields’ classification system for hereditary dental malformations, three subgroups of dentinogenesis imperfecta (DGI type I, DGI type II, and DGI type III) and two types of dentin dysplasia (DD type I and II) were classified to describe their different clinical features [1]. Type I Dentin Dysplasia, also named DD-I or DTDP1 or Radicular Dentin Dysplasia (ORPHA:99789; ICD-10: K005), is characterized by short, conical, pointed roots or rootless teeth and affects both primary and permanent dentition [1,2,3]. The prevalence of DD-I is estimated at 1/100,000. According to Shields’ classification, type II dentin dysplasia (DD-II) is characterized by primary teeth with brown or blue opalescent color [1]. Ciola added a third group of dentin dysplasia (type III dentin dysplasia or DD-III) that includes teeth affected by the combination of types I and II dentin dysplasia [4]. Later, Carroll et al. suggested updating the classification of dentin dysplasia by proposing four subgroups in DD-I called a, b, c, and d to better distinguish the different forms in which this dentin defect manifests and differentiates from DD-II [5,6].

The first differential diagnosis for DD-I is DD-II. Dentin dysplasia type II demonstrates numerous features of DD-I, but in contrast to DD Type I, root lengths are normal in both dentitions. A radiographic finding of a thistle-tube-shaped pulp chamber in a single-rooted tooth increases the likelihood of DD-II diagnosis [7]. The association of periapical radiolucencies with noncarious teeth and without obvious cause is an important characteristic of DD-I. In DD-I patients, teeth usually appear normal in shape and color [5,6,8]. However, the roots are pointed and, on radiological examination, are characterized by apical and conical constrictions [6,8]. Teeth are usually mobile; they often undergo the formation of abscesses, granulomas, and cysts and can fall out prematurely [9,10,11]. The pulp chambers do not fill in before eruption in DD-II, while the aberrant formation of dentin can cause partial or total obliteration of the pulp in DD-I [12,13,14].

Further differential diagnoses are posed with diseases that have radiological and/or clinical signs similar to DD-I, including the different types of dentinogenesis imperfecta and with the following conditions causing premature tooth loss: Kostmann syndrome; cyclic neutropenia; Chediak–Hegashi syndrome; cell histiocytosis Langerhans syndrome; Papillon–Lefèvre syndrome; hypophosphatasia; and vitamin D-resistant rickets [6,7,11,13,15,16,17]. DD-I is caused by the upregulation or downregulation of many genes involved in odontogenesis (Dspp, Dmp1, Runx2, Pax9, Bmp2, Dlx2, vPS4B, Ssuh2, and SMOC2) [13,14,15,17,18,19,20,21,22,23]. The most documented forms of DD-I are those related to the mutation of the Dspp gene (4q21.3), which encodes dentin sialophosphoprotein, a precursor of dentin sialoprotein and dentin phosphoprotein, which are involved in dentinogenesis [18,19,20,21,22]. The diagnosis of DD-I is essentially based on the evidence collected in radiological investigations such as orthopantomography (abnormal roots, pulp obliteration, partially obliterated crescent-shaped pulp chamber, and, rarely, pulp stones). Molecular tests can be used to support the diagnosis. DD-I is inherited in an autosomal dominant or, more rarely, recessive manner. A person with the disease has a 50% chance that his children will inherit the disease [10,11,14,20,24]. The age of the patient at the time of diagnosis greatly influences the dental prognosis; however, the severity of dysplasia does not always allow for planning a conservative approach despite an early diagnosis [6,7,9,13,19,24,25].

Currently, the only guidelines available are those established by the American Academy of Pediatric Dentistry (AAPD), which were updated in 2013 [26]. The American Academy of Pediatric Dentistry (AAPD) recommends as general consideration and management principles the maintenance of teeth for as long as possible as the only goal to be set given the inevitably poor prognosis. Preventive care focuses on meticulous hygiene, as shortened roots predispose to the onset of dental loss due to periodontitis. The typically unfavorable crown/root ratio means that feasible restorative treatments can be only prosthetics (dentures, overdentures, partial dentures, and/or dental implants). Endodontic therapy, both orthograde and retrograde, must be considered with caution because in teeth with extremely short roots, the failure rate, even in the short term, is very high [25,26,27].

In the literature, three previous case reports with the literature review are useful to clinically describe this hereditary condition [28,29,30]. However, they do not provide a general description or a specification (for the cases presented) of the genetic diagnosis. In this paper, the cases of two siblings with DD-I and the oral and radiographic manifestations of this rare condition are described. Genetic testing confirms DD-I. Despite the first dental visit occurring at an early age, the permanent teeth of the older of the two brothers, who had already undergone orthodontic therapy with a rapid palate expander, are already significantly compromised. Despite receiving dental therapies (and dental X-rays) for mouth rehabilitation for teeth lost over time, the father of these two children had never been informed by any other dentist of a diagnostic suspicion for this or other diseases.

## 2. Case Presentation

DD-I was diagnosed in the proband (patient A), in his sister (patient B), and their father. The youngest sister was unaffected. Pedigree data were consistent with autosomal dominant inheritance and indicated a probable presence of the disease in the paternal grandfather and paternal aunt (Figure 1).

### 2.1. Patient A

A 12 years old Caucasian male patient was referred by his pediatrician for an orthodontic checkup in the dental office of one of the authors for an assessment of the need for orthodontic therapy. In the intra-oral examination, the patient presents a state of poor oral hygiene for almost all of the permanent dentition. The upper lateral right incisor was in a crossbite. Moderately unusual mobility of the upper lateral incisors and first lower premolars were observed. The medical history was negative for trauma, and the parents reported only signs indicating previous dental abscesses for which they did not seek any dental treatment. The parents reported that they considered the abscesses insignificant events attributed to the loss of primary teeth (Figure 2). The parents also reported that the child, two years before this check-up, had undergone rapid palate expander therapy with another specialist for 1 year, which was completed pending therapeutic reevaluation following dental changes and skeletal development. Given the mobility of the permanent teeth and the presence of destructive caries on the first upper deciduous molar, it was decided to perform orthopantomography. The radiographic appearance immediately raised the diagnostic suspicion of DD-I, as the roots of the permanent teeth were completely short or absent and showed typical sharp conical apical constrictions and crescent-shaped pulpal remnants parallel to the cementoenamel junction (Figure 3). However, the rootlessness of upper lateral incisors could be caused by the ectopic eruption of the upper canines. From the observation of two previous dental radiographs taken when the patient was 7 and 10 years old, dentinal dysplasia affected both the deciduous and permanent dentition of our patient (Figure 4 and Figure 5). DD-I was also evident during the orthodontic treatment (Figure 4).

### 2.2. Patient B

To extend the familial diagnosis, the sister of the patient was also subjected to an intraoral examination (Figure 6), and an orthopantomography obtained one year before for a previous dental check-up with another dentist who had not raised any diagnostic suspicion for dentinal anomalies was observed. The girl, 10 years old, radiographically presented the characteristic signs of type I dentinal dysplasia as well (Figure 7). Her teeth did not show mobility or signs of alteration in their coronal external structure. The only exception was the anomalous position of the two upper right incisors and the contracted appearance of the upper dental arch.

### 2.3. Father

The family history, initially negative for DD-I, was refuted by the observation of an orthopantomography of the children’s father, who had only reported a history of loss of some dental elements, which were later replaced by prosthetics (Figure 8). The family had never before been informed by any other dentist of a diagnostic suspicion for this or other genetic diseases involving the teeth.

### 2.4. Other Family Members

The father of the two children reported that his mother (deceased) had never had dental problems but that both his father (deceased) and his only sister (who lives in another country and has no children) had a history of the loss of many permanent dental elements during their lives (the sister recalled that the loss of dental elements has occurred since she was very young). The patients’ youngest sister (4 years old) has an apparently normal deciduous dentition, but due to her age and the absence of clinical signs, no dental X-rays were required. Therefore, we suspected transmission of the disease by the grandfather of the children to their father and aunt (Figure 1).

### 2.5. Differential Diagnosis and Genetic Tests

The diagnosis was based on history, clinical examination, and radiographic features. Dentin dysplasia type II (DD-II) was excluded. In this condition, the primary dentition has features resembling the dentinogenesis imperfecta type II (DGI-II), and the permanent dentition is unaffected or shows mild radiographic abnormalities such as thistle-tube deformity of pulp chambers and pulp stones. Dentinogenesis imperfecta type I (DG-I) always occurs in association with osteogenesis imperfecta. The radiographic features of DG-I (short and constricted roots and pulpal obliteration) are not sufficiently consistent with the clinical presentations of these patients because the teeth of both dentitions were not typically translucent or amber. Dentinogenesis imperfecta type II (DGI-II) is a severe form of dentinogenesis imperfecta that can affect both primary and permanent dentitions. In contrast to the presentation of our patients, in DGI-II, teeth can appear weak and discolored, and their crowns are bulbous and have marked cervical constrictions. Radiographically, DGI-II exhibits opacifications of dental pulps, short roots, and bell-shaped crowns. Dentinogenesis imperfecta type III (DGI-III), similarly to DD-I, is associated with multiple periapical radiolucencies in noncarious teeth, but the teeth in DD-I, as in our patients, are clinically normal. All patients underwent specific genetic tests (sequencing with the Sanger method) for rare odontological diseases, which confirmed the diagnosis. DD-I is associated with the deficiency of a protein-coding gene called SMOC2 (SPARC Related Modular Calcium Binding 2) with autosomal dominant inheritance. The single nucleotide variant NM_001166412.2(SMOC2): c.84+1 G>T located on 6q27 was identified. The PCR and Sanger sequencing confirmed the mutation in the two affected children and in their father, while the younger sister (4 years old) was not affected by the same mutation (Figure 1).

### 2.6. Dental Management of the Affected Children and Father

Following the diagnosis, the patients began a process of close periodic checks every 3–4 months to monitor their situation and check the conditions of oral hygiene, which were more necessary than ever to avoid periodontal inflammation and carious lesions. Patients and parents were also emotionally and psychologically supported in the need to plan the interventions that became necessary once they were of suitable age to replace their missing teeth with dental implants. After a few months during the COVID-19 pandemic, the compromised situation of patient A led to the spontaneous loss of the permanent lateral incisors, which were replaced by a light nylon removable prosthesis with aesthetic hooks (Figure 9), which had high aesthetic value and excellent function due to the flexibility of the material. Written informed consent from the parents was requested and obtained, as was the permission to disclose intraoral and extraoral photos and radiographs with respect to the privacy and anonymization of personal data.

## 3. Discussion

The case of these family members demonstrates how DD-I can go undiagnosed for a long time and even be ignored if its unequivocal radiographic signs are not recognized. The importance of the early diagnosis of rare odontological diseases is evident because it is the key to preventing and controlling the negative effects of these diseases, which affect the oral health and quality of life of affected individuals [10,11,19]. In the literature, there are other reports of late family diagnoses with important manifestations of DD-I in pediatric patients that have captured the parents’ attention because their children have become symptomatic (hypermobility on permanent elements, abscesses, and dental pain), from which the diagnosis was then made for one of the parents who had been treated for premature loss of permanent teeth with prosthetics and for whom the diagnostic suspicion of rare odontological diseases had not even been raised [9,18,20,25]. The etiology of DD-I remains unclear [10,15,18,19,26,27,28,29,30]. The hypotheses proposed over the years differ, but according to the etiology currently considered most likely, the abnormal root morphology is caused by abnormal differentiation and/or function of the odontoblasts that form dentin. The odontoblasts are derived from the epithelial cells of the Hertwig sheath and start their differentiation after having detached and migrated into the dental papilla [9,20,21]. Dental management of patients with DD-I poses several problems [10,12,13,18,19,24,25,26,31,32]. When pulp necrosis and periapical abscess occur, extraction is suggested as a treatment [10,12,26,28,29]. For this reason, patients with DD-I in permanent dentition need to be monitored regularly for any abscess/apical lesion [26,31,32]. The therapeutic approach to DD-I is as conservative as possible to preserve an already very vulnerable dentition [4,18,20]. Endodontic treatment of teeth affected by DD-I is not appropriate due to the complexity of root canals or to the completely altered internal structure of the pulp [9,25,29,31]. Peri-apical surgery and retrograde filling are recommended in teeth with long roots [25]. Orthodontic treatment is suggested but with caution. In fact, further root resorption, loosening of teeth, and premature exfoliation may occur due to the resistance of the short roots to the orthodontic forces [1,20,32,33]. In patient A, we cannot exclude that the orthodontic treatment could have contributed to accelerating the root resorption of the teeth of the upper arch. A correct diagnosis before orthodontic treatment would perhaps have allowed for preserving the lost teeth and the root structures of the other elements for a longer time. External root resorption (EER) often occurs in subjects treated with rapid maxillary expansion. The first molars and the upper incisors are the teeth most affected by the resorption process, with a significant reduction in radicular volume and length [34,35]. The choice to use a bone-borne expansion and not traditional tooth-borne expansion, as in this patient, does not significantly reduce this risk [34]. According to recent data on cone beam computed tomography (CBCT), the volume of the pulp chamber of posterior teeth, especially at the level of the horns, is also affected by a volumetric reduction with any type of rapid expander [36,37]. These data must be taken into account if this pathology is recognized before starting orthodontic therapy, even when an expansion of the upper arch is necessary. Early tooth exfoliation promotes maxillomandibular bone atrophy, so treatment with a combination of onlay bone graft and sinus lift may be required to achieve implant rehabilitation [19]. The main goal in the dental management of this disorder remains to preserve the natural teeth for as long as possible to preserve the bone. For this reason, in children, in particular, strict oral hygiene measures and dietary instructions must be established and maintained, as in other syndromic situations affecting teeth [30,32]. It should be noted that the loss of permanent upper lateral incisors (in an advanced state of compromise already at the time of our first observation) in the older brother occurred during the first wave of the COVID-19 pandemic. The delayed prosthetic rehabilitation had an important psychological impact on the child, although, in that period, many remote-control strategies were implemented not to make pediatric patients feel neglected [38,39,40,41]. The dentist has a fundamental role in the early diagnosis of this disorder and in guiding patients in the choice of measures to prolong the retention of the affected teeth [33,42,43,44]. These cases demonstrate the importance of pediatric dental and orthodontic examination, as some rare genetic conditions such as DD-I may remain unknown and undiagnosed or misdiagnosed until problems such as dental abscesses or the tooth mobility of the permanent teeth arise in the absence of trauma or other factors. The diagnostic suspicion that is confirmed only by genetic research can be raised by the knowledge of this and of similar rare oral conditions and their clinical manifestation and radiographic aspects of the deciduous and permanent teeth [10,24,26]. In this regard, the cases presented offer a genetic connotation different from the most widespread ones. DD-I is frequently described as a condition that follows an autosomal dominant pattern of inheritance caused by mutations in the DSSP genes [10,14,19,20]. The mutated genes SSUH2-SSU-2 homolog and vPS4B-vascular protein sorting 4 homolog B are responsible for DD-I, too [14,19,20,21,22]. The mutation of the SPARC-related modular calcium binding 2 gene, called SMOC2 (located on 6q27chromosome), is associated with an atypical dentin dysplasia due to SMOC2 deficiency. This subtype of DD-I, which follows an autosomal recessive inheritance, is characterized by extreme microdontia, oligodontia, abnormal tooth shape, short roots, enamel hypoplasia, and anterior open bite [13]. Our patients were found to carry a SMOC2 mutation with a phenotype suggestive of an autosomal dominant inheritance showing the wide yet unknown genetic heterogeneity of this disease [15,18,19,24,26]. Orthodontic treatment in persons affected by this kind of condition based on the genetic heritage (root length reduced, decreased crown/root ratio, and premature loss) can be highly counterproductive, as can endodontic treatment [33,45]. For this reason, it is extremely important for the oral health of the growing affected individual to make a diagnosis early and to involve other family members who can be equally affected by this condition and have been ignored by dentists by attributing the premature loss of dental elements to other causes, as in the case of the father presented here. Although no similar cases are described in the literature, other cases of arrested root formation underline the importance of always considering an alarming signal of the lack or malformations of dental elements in the family members of a patient who has abnormal and insufficient root development [24,46]. In fact, the authors in one of these studies state, “Arrested root development is difficult to predict, but a potential warning sign is a family history of malformed or missing teeth. Proper, adequate, and accurate records continue to remain critical for both medical and legal purposes in the treatment of patients with potential problems in root agenesis” [47]. We agree with this statement, which is also applicable to our case, in that the diagnosis of the children’s father could have led to a further early diagnosis for the children, given the 50% probability that the children of an affected individual will present the same pathology. This may have allowed for better management of the father’s oral health, which indirectly would have been preventive for the children’s health, even by visiting a non-pediatric dentist or orthodontist. As rightly observed by other authors, when a dentist, general practitioner, or specialist is faced with a rare condition and ignores it due to his ignorance or negligence in performing a thorough diagnostic workup, the lack of diagnosis cannot be attributed to others and the clinician must be considered responsible not only ethically but also legally [47,48,49]. Diagnostic errors are a known problem in health care practice, but even if data on diagnostic errors in the dental field are lacking [48], it is extremely important to report dental diagnostic errors and misdiagnosed dental conditions, such as the case of the father reported here, for whom the lack of diagnosis was not the only failure, as the dentistry profession, the members of which sometimes present themselves as too self-confident and superficial in taking care of patients, was also affected.

## 4. Conclusions

In the field of rare dental diseases, the early diagnosis of the various forms of dentin dysplasia is extremely important. The ability to recognize the distinctive radiographic signs of less clinically evident forms, such as DD-I, in which dental crowns appear normal, can be very important to guide the patient and their family members from childhood. In the presence of permanent dental elements that are unusually mobile, with or without abscess or inflammatory phenomena, dental orthopantomography should always be carefully evaluated, especially if one of the parents reports that they themselves have lost healthy permanent dental elements from an early age for no known reason.

## Figures and Tables

**Figure 1 medicina-59-01477-f001:**
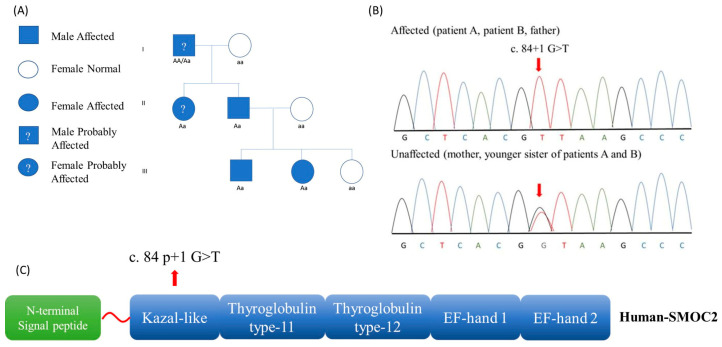
(**A**). Pedigree analysis suggested an autosomal dominant inheritance. (**B**). Sanger sequencing chromatograms; red arrow indicates the position of mutation c.84+1 G>T. (**C**). Schematic illustration of the key functional domains and related mutation reported in these cases.

**Figure 2 medicina-59-01477-f002:**
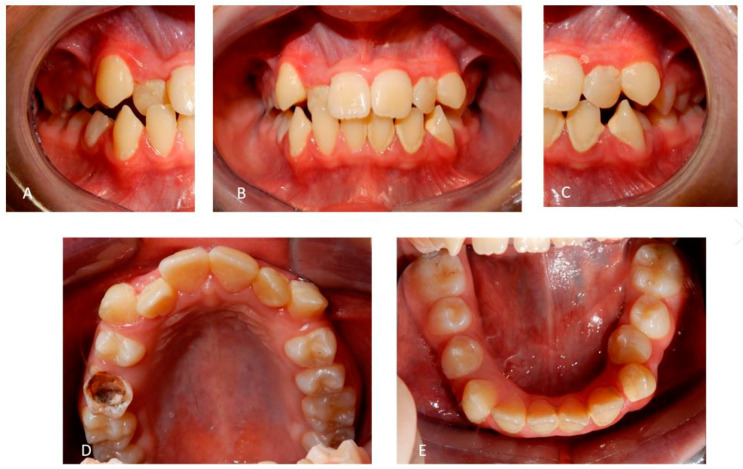
(**A**) Cross-bite of the right upper lateral incisor. (**B**) Front view showing malocclusion and poor oral hygiene. (**C**) Malocclusion observed on the left side of the patient. (**D**) Occlusal view of the upper arch showing destructive caries on the right second deciduous molar. (**E**) Occlusal view of the lower arch showing an abnormal position of the left canine and second premolar. Morphology and color of teeth appear normal. In DD-I, occasionally, teeth appear slightly amber or have bluish–brown shine (in primary teeth).

**Figure 3 medicina-59-01477-f003:**
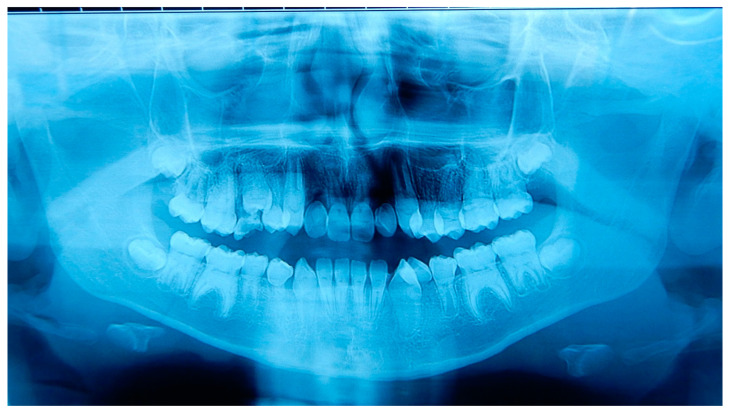
Orthopantomography of patient A. This radiograph was made at the moment of the dental check-up of patient A (12 years old), showing the typical abnormalities of the DD-I affect the roots (short with apical constrictions) and the pulp (pulp chamber and canals obliteration, half-moon shaped pulp chamber remnants) and the rootless permanent teeth (the upper lateral incisors and the lower first premolars).

**Figure 4 medicina-59-01477-f004:**
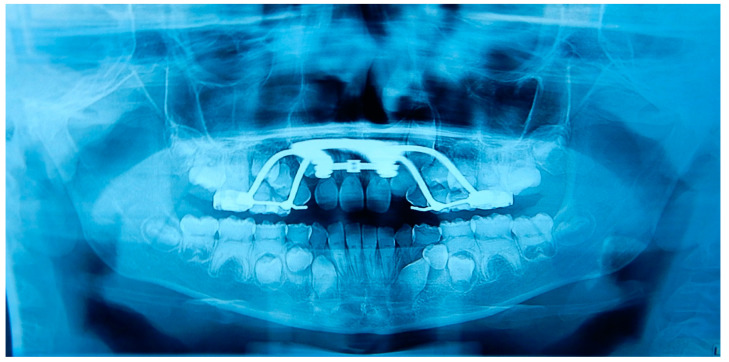
Previous orthopantomography of patient A. This radiograph was made when patient A was 10 years old, during orthodontic therapy with a rapid palatal expander. The abnormal development of dentition was evident. The permanent teeth have short roots and half-moon-shaped pulp chambers.

**Figure 5 medicina-59-01477-f005:**
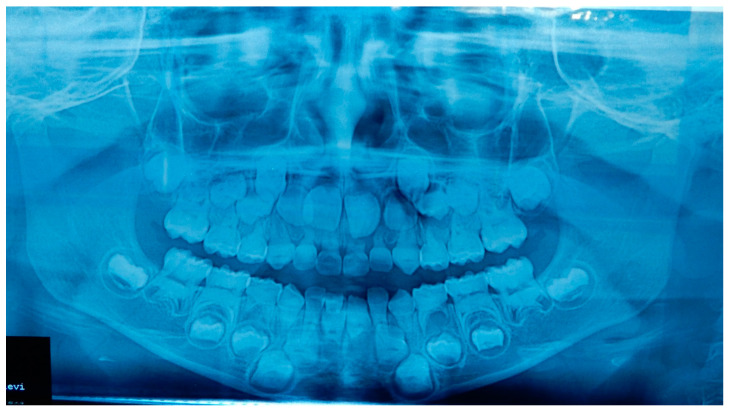
Previous orthopanthomography of patient A. This radiograph was made when patient A was 7 years old, showing the abnormal development of both the dentitions, deciduous and permanent: short roots and pulp chamber and canals obliteration.

**Figure 6 medicina-59-01477-f006:**
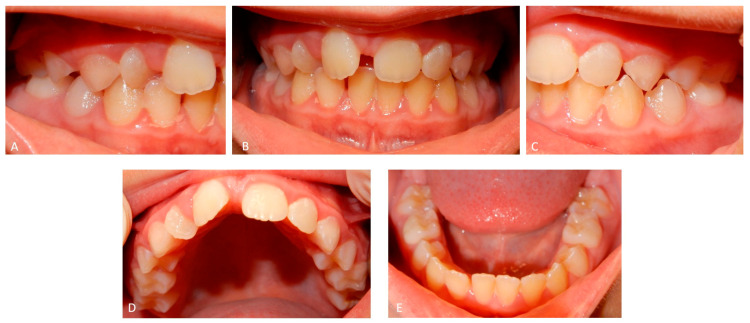
Intra-oral view of the dentition of patient B. (**A**) The right upper lateral incisor is rotated and in cross-bite with the lower incisor. (**B**) The front view shows the wide upper diastema between the central incisors. (**C**) The left side showing a correct occlusion. (**D**) Occlusal view of the upper arch showing rotations of the right lateral and central incisors. (**E**) Occlusal view of the lower arch showing teeth apparently free of any abnormality. Both deciduous and permanent teeth appear normal in morphology and color.

**Figure 7 medicina-59-01477-f007:**
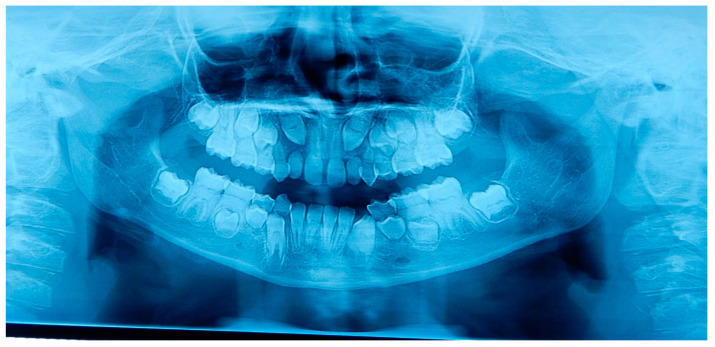
Orthopantomography of patient B. This radiograph was made almost one year before our observation for a dental check-up with another dentist. The typical abnormalities of the DD-I (shorter than normal roots and pulp chamber and canals obliteration) affecting both deciduous and permanent teeth are visible, especially on permanent upper canines, lateral incisors, lower incisors, and molars and on deciduous lower molars.

**Figure 8 medicina-59-01477-f008:**
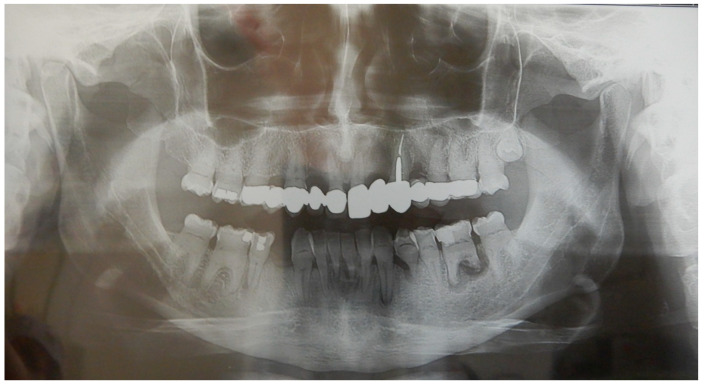
Orthopantomography of the children’s father. This radiograph shows the typical radicular and pulpal features of the DD-I observed in his children (pulp chamber and canals obliteration and short roots). There are also abscesses on the left lower canine, first premolar, first molar, and two upper fixed prostheses on natural teeth extended from first molar to central ipsilateral incisor with intra-radicular pin on one of the canines treated endodontically.

**Figure 9 medicina-59-01477-f009:**
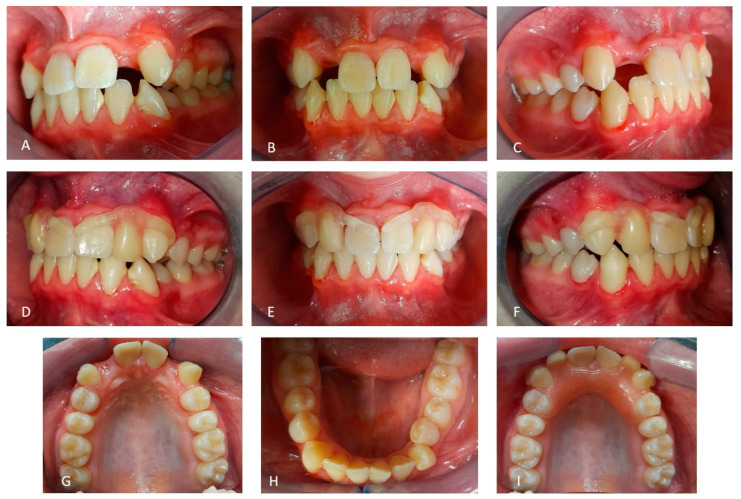
Results of intra-oral view of the dentition of the patient after the loss of his upper lateral incisors occurred during the first outbreak of SARS-Cov2 pandemic. (**A**–**C**,**G**) The lateral, front, and upper occlusal views show the absence of the upper lateral incisors. (**D**–**F**,**I**) The lateral and front views of the mouth rehabilitated with a removable prosthesis in the upper arch. (**H**) In the lower arch, teeth are still present, but during our last periodic examination, the left first premolar started to show an increased mobility.

## Data Availability

Not applicable.

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
