# Peer review of "Type I Dentin Dysplasia: The Literature Review and Case Report of a Family Affected by Misrecognition and Late Diagnosis"

_medicina, 2023, doi:10.3390/medicina59081477_

Round 1

Reviewer 1 Report

Type I dentin dysplasia: literature review and case report of a family 2 affected by misrecognition and late diagnosis.

This study case report manuscript investigates the rare condition of dentin dysplasia in a family with two siblings and one of the parents.

The introduction is well written, comprehensive and accommodates the reader with the main features of the subject.

The case presentation supplies a coherent and proper description of the clinical conditions of the patients. The differential diagnosis is comprehensive and clearly written. The treatment plan is logical, and the reader understands the aim of the treatment of this condition.

The discussion part supplies further data for the clinician explaining the genetic origins of the condition and the main direction of the treatment plan, as well as the importance of periodical check-ups.

The conclusions are focused on the importance of identifying an early diagnosis of these rare conditions.

The references support the data from this manuscript.

Overall, the manuscript is well written, comprehensive and of a particular significance for the clinician and could be published.   

Author Response

Comments and Suggestion for Authors REVIEWER 1

This study case report manuscript investigates the rare condition of dentin dysplasia in a family with two siblings and one of the parents.

The introduction is well written, comprehensive and accommodates the reader with the main features of the subject.

The case presentation supplies a coherent and proper description of the clinical conditions of the patients. The differential diagnosis is comprehensive and clearly written. The treatment plan is logical, and the reader understands the aim of the treatment of this condition.

The discussion part supplies further data for the clinician explaining the genetic origins of the condition and the main direction of the treatment plan, as well as the importance of periodical check-ups.

The conclusions are focused on the importance of identifying an early diagnosis of these rare conditions.

The references support the data from this manuscript.

Overall, the manuscript is well written, comprehensive and of a particular significance for the clinician and could be published.  

Author’s Reply to the Review Report (Reviewer 1)

Dear Reviewer,

thanks for your overall positive evaluation. We provided some change to the original version of the manuscript in order to satisfy the comments and suggestions of the Reviewer 2. They are highlighted in yellow colour throughout the text. We hope these changes confirm your opinion.

Sincerely,

the Corresponding Author and co-Authors

Reviewer 2 Report

the title of this manuscript including literature review, however the finding from the literature review is too short and not comprehensive. Please add more info so that it is suit the tittle claim. 

Figure 1 not so clear, please provide a new figure with a good quality photo, clear and sharp.

figure 6, please crop unnessasary area with is not related to the tooth/teeth

Author Response

Comments and Suggestions for Authors – REVIEWER 2

  1. The title of this manuscript including literature review, however the finding from the literature review is too short and not comprehensive. Please add more info so that it is suit the tittle claim.

  1. Figure 1 not so clear, please provide a new figure with a good quality photo, clear and sharp.

  1. figure 6, please crop unnessasary area with is not related to the tooth/teeth

Authors’ Reply to the Review Report (Reviewer 2)

Dear Reviewer,

thank you for your evaluation and for providing us with suggestions for improvement for our manuscript. We have tried to satisfy your comments and highlighted them in the yellow text. You won't find any other changes because Reviewer 1 has stated that there are not changes needed. We hope that the updates made are adequate to your valuable comments.

  1. The title of this manuscript including literature review, however the finding from the literature review is too short and not comprehensive. Please add more info so that it is suit the tittle claim.

Consulting the most recent articles published in this journal in the formula "case report and literature review" [1-8], we do not seem to have omitted a description of what is present in the literature about our topic, however, we have taken your suggestion to check the introduction and understand if further contents could add relevance to the title and contents.

A case series with review of the literature titled “Dentin Dysplasia Type I: case series and review of the literature” (published in 2015 in the International Journal of Dental and Health Sciences) introduces its manuscript with a short paragraph and 10 references. In the discussion 8 references are added for a total of 18 references [9].

A previous case report titled “Dentin dysplasia type I: a case report and review of the literature” (published in 2010 in the Journal of Medical Case Reports) describe in its introduction the DD-I citing just 6 references and adding just 10 references in the discussion [10]. In our manuscript, our introduction provide a wide description of history, classifications, etiopathogenesis, differential diagnosis and diagnosis, guidelines with a total of 27 references (in the last version 30). In our discussion we provide a total number of further 20 references equal to a total of 46 (in the last version 49).

An older case report with review of the literature, published in 1997, in the Journal of Dentistry for Children, counts in its introduction with literature review 15 references. Further 19 references are discussed in the related section for a total of 34 references [11].

However, this verification was fundamental, as well as for our further study on the subject, to add these three bibliographic entries to our introduction (page 1, lines 86-88; page 14, lines 457-462) as references to previous case reports that provided data and review of the existing literature lack of genetic diagnosis. So thanks again for this tip.

  • Vâţă, A.; Irimie-Băluţă, E.; Roşu, F.M.; Onofrei, I.M.; Loghin, I.I.; Perţea, M.; Avădanei, A.N.; Miron, M.; Rădulescu, L.; Eşanu, I.; et al. Polymicrobial Bacterial Meningitis in a Patient with Chronic Suppurative Otitis Media: Case Report and Literature Review. Medicina 2023, 59, 1428. https://doi.org/10.3390/medicina59081428
  • Zhou, L.; Huang, J.; Li, H.; Duan, H.; Hua, Y.; Guo, Y.; Zhou, K.; Li, Y. Impaired Cardiomyocyte Maturation Leading to DCM: A Case Report and Literature Review. Medicina 2023, 59, 1158. https://doi.org/10.3390/medicina59061158
  • Bužinskienė, D.; Marčiukaitytė, R.; Šidlovska, E.; Rudaitis, V. Ovarian Leydig Cell Tumor and Ovarian Hyperthecosis in a Postmenopausal Woman: A Case Report and Literature Review. Medicina 2023, 59, 1097. https://doi.org/10.3390/medicina59061097
  • Park, S.-D.; Kim, M.-S.; Han, M.-H.; Kim, Y.-J.; Jung, H.-Y.; Choi, J.-Y.; Cho, J.-H.; Park, S.-H.; Kim, C.-D.; Kim, Y.-L.; et al. Renal Sarcoidosis-like Reaction Induced by PD-1 Inhibitor Treatment in Non-Small Cell Lung Cancer: A Case Report and Literature Review. Medicina 2023, 59, 991. https://doi.org/10.3390/medicina59050991
  • Ciongariu, A.-M.; Dumitru, A.-V.; Cîrstoiu, C.; Crețu, B.; Sajin, M.; Țăpoi, D.-A.; Ciobănoiu, A.-D.; Bejenariu, A.; Marin, A.; Costache, M. The Conundrum of Dedifferentiation in a Liposarcoma at a Peculiar Location: A Case Report and Literature Review. Medicina 2023, 59, 967. https://doi.org/10.3390/medicina59050967
  • Hsiao, P.-H.; Lin, E.-T.; Chen, H.-T.; Lo, Y.-S. Complete Intradural Interbody Cage Migration in Lumbar Spine Surgery: A Case Report and Literature Review. Medicina 2023, 59, 956. https://doi.org/10.3390/medicina59050956
  • Matei, S.-C.; Dumitru, C.Ș.; Oprițoiu, A.-I.; Marian, L.; Murariu, M.-S.; Olariu, S. Female Gonadal Venous Insufficiency in a Clinical Presentation Which Suggested an Acute Abdomen—A Case Report and Literature Review. Medicina 2023, 59, 884. https://doi.org/10.3390/medicina59050884
  • Iordache, S.; Cursaru, A.; Serban, B.; Costache, M.; Spiridonica, R.; Cretu, B.; Cirstoiu, C. Melorheostosis: A Review of the Literature and a Case Report. Medicina 2023, 59, 869. https://doi.org/10.3390/medicina59050869
  • Kumar, N., Ansari, S., Shrivastava, H., et al. Dentin Dysplasia Type I: case series and review of the literature. Int J Dent Health Sci 2015; 2(1): 198-202.
  • Toomarian, L., Mashhadiabbas, F., Mirkarimi, M. et al. Dentin dysplasia type I: a case report and review of the literature. J Med Case Reports 4, 1 (2010). https://doi.org/10.1186/1752-1947-4-1.
  • Ansari G, Reid JS. Dentinal dysplasia type I: review of the literature and report of a family. ASDC J Dent Child. 1997;64(6):429-434.

  1. Figure 1 not so clear, please provide a new figure with a good quality photo, clear and sharp.

We have changed the image as requested (page 5, line 188). Sanger sequencing chromatograms (1B) can’t be modified since it is an original image of the genetic documentation provided to us.

  1. figure 6, please crop unnessasary area with is not related to the tooth/teeth.

We have changed the image as requested (page 7, line 219). The occlusal view of the upper arch (6D) can’t be furtherly cropped. The little young patient was not very cooperative during the photo session and refused mirror and cheek retractors. It's the best photo we have.

Sincerely,

the Corresponding Author and Co-Authors
